# Targeting HER2 in Gastroesophageal Adenocarcinoma: Molecular Features and Updates in Clinical Practice

**DOI:** 10.3390/ijms25073876

**Published:** 2024-03-30

**Authors:** Maria Bonomi, Daniele Spada, Gian Luca Baiocchi, Andrea Celotti, Matteo Brighenti, Giulia Grizzi

**Affiliations:** 1Oncology Unit, ASST Cremona, 26100 Cremona, Italy; maria.bonomi@asst-cremona.it (M.B.); daniele.spada@asst-cremona.it (D.S.); matteo.brighenti@asst-cremona.it (M.B.); 2Department of Surgery, ASST Cremona, 26100 Cremona, Italy; gianluca.baiocchi@asst-cremona.it (G.L.B.); andrea.celotti@asst-cremona.it (A.C.)

**Keywords:** gastroesophageal adenocarcinoma, HER2, immunotherapy, trastuzumab, trastuzumab deruxtecan, ct-DNA, heterogeneity

## Abstract

Gastroesophageal adenocarcinoma (GEA) is one of the principal causes of death related to cancer globally. Human epidermal growth factor receptor 2 (HER2) is a tyrosine kinase receptor which is found to be overexpressed or amplified in approximately 20% of GEA cases. In GEA, the identification of HER2-positive status is crucial to activate a specific anti-HER2 targeted therapy. The landmark ToGA trial demonstrated the superiority of adding trastuzumab to platinum-based chemotherapy, becoming the first-line standard of treatment. However, unlike breast cancer, the efficacy of other anti-HER2 drugs, such as lapatinib, pertuzumab, and T-DM1, has failed to improve outcomes in advanced and locally advanced resectable GEA. Recently, the combination of trastuzumab with pembrolizumab, along with chemotherapy, and the development of trastuzumab deruxtecan, with its specific bystander activity, demonstrated improved outcomes, renewing attention in the treatment of this disease. This review will summarise historical and emerging therapies for the treatment of HER2-positive GEA, with a section dedicated to the HER2 molecular pathway and the use of novel blood biomarkers, such as circulating tumour DNA and circulating tumour cells, which may be helpful in the future to guide treatment decisions.

## 1. Introduction

Gastric adenocarcinoma and gastroesophageal junction adenocarcinoma (hereafter, GEA), with more than one million estimated new diagnoses per year, are the fourth cause of cancer-related death and the fifth-most diagnosed tumours, representing a challenge for clinicians worldwide [1]. Several patients have unresectable tumours at the time of diagnosis or experience a recurrence of the disease after undergoing surgery with curative intent [2]. 

Standard chemotherapy for advanced GEA yields response rates of 20–40% and a median overall survival (mOS) of only 10–12 months [3]. In the era of precision medicine and its ever-growing utilisation in clinical practice, molecular targeted therapies have achieved increasing attention for the treatment of GEA [4,5]. Among these, alterations in human epidermal growth factor receptor 2 (HER-2) have the clearest clinical significance and are frequently identified across various cancer types. In GEA, overexpression/amplification of HER2 is identified in about 20% of cases [6,7]. HER2 was the first recognised treatment target in GEA, based on the results of adding trastuzumab to platinum-based chemotherapy in the Trastuzumab for Gastric Cancer (ToGA) trial for the treatment of HER2-positive advanced disease [7]. Moreover, this trial was fundamental in identifying and standardising the definition of HER2-positive GEA, defined as an immunohistochemistry (IHC) score 3+ or fluorescence in situ hybridisation (FISH) HER2–CEP17 ratio ≥ 2. 

GEA is a complex disease, and the two traditional morphological classifications by Lauren and the World Health Organisation (WHO) are unable to include the molecular heterogeneity of this tumour. Since 2014, the Cancer Genome Atlas (TCGA) research network has proposed a new classification of GEA, divided into four molecularly distinct subtypes: Epstein–Barr virus positive (EBV+), microsatellite instable (MSI), genomically stable (GS), and chromosomal unstable (CIN) [8]. In the TCGA classification, a specific “HER2-enrich” subtype is not categorised and HER2 amplification is associated with both the CIN and EBV subgroups [9]. These data require reflection on the heterogeneous expression of HER2 in GEA which might not be an independent oncogenic driver to block the HER2-mediated signalling pathway. 

In this review, we will summarise historical clinical studies and recent innovations in the treatment of HER2-positive metastatic and locally advanced resectable GEA. Additionally, we will discuss future therapeutic strategies and the development of new anti-HER-2 drugs for this disease. 

## 2. HER2 in Gastroesophageal Adenocarcinoma: The Molecular Pathway

*ERBB2* is a proto-oncogene localised on chromosome 17q21, encoding the HER2 receptor, also known as Neu or ErbB2. HER2 is a member of the HER receptor family, which includes EGFR (epidermal growth factor receptor)/HER1, HER2/neu, HER3, and HER4. The protein structure of all HER receptors consists of a tyrosine kinase intracellular domain, a transmembrane lipophilic domain, and a cysteine-rich extracellular ligand-binding domain [10]. The HER receptors are able to homodimerize or to heterodimerize with other members of the HER family [11]. HER2 is a transmembrane tyrosine kinase receptor that plays a crucial role in triggering multiple downstream signalling pathways, such as RAS/RAF/MAPK and PI3K/AKT, leading to cell growth and differentiation [12,13] (Figure 1). The extracellular domain of HER2 is composed of four subdomains, including the dimerisation domain II, which is essential for ligand-induced heterodimerization. HER2 has no activating ligand but is the preferred dimerisation partner for all other HER receptors (HER1 and HER3 in particular) because it is in an open conformation and is continuously available for dimerisation [11,14,15].

Ligand-independent activation of HER2 may result from specific mutations in the receptor or from HER2 overexpression [10]. Overexpression and amplification of HER2 were initially identified in breast cancer and are associated with poor outcomes in this cancer subtype [16]. Many studies have shown that HER2 alterations are also present in other malignancies, including colorectal cancer, lung cancer, ovarian cancer, prostate cancer, and GEA [17,18]. 

According to different studies, HER2 amplification rates in GEA range from 12 to 27%, while HER2 overexpression ranges from 9% to 23% [19,20,21,22,23].

In GEA, HER2 is more common in the gastroesophageal junction tumour (30% vs. 20% of the mid and distal stomach) and in the intestinal subtype (32.2% vs. 6.1% of the diffuse type) [24].

HER2-positive tumours typically exhibit a higher tumour grade and a capability to grow and spread more rapidly than cancers with normal HER2 expression. Unlike breast cancer, where HER2-positive disease is associated with poor outcomes, the prognostic role of HER2 overexpression in metastatic or resectable GEA has been considered controversial [25]. In 2012, a systematic review of forty-two publications was conducted to address this issue [19]. A large part of the studies (71%) demonstrated poor survival and a pattern of tumour progression, suggesting that overexpression or amplification of HER2 is a negative prognostic factor.

At the time of diagnosis, HER2 testing is recommended for all patients [26,27]. Compared to breast cancer, GEA exhibits unique immunostaining characteristics, with a more heterogenous basolateral membrane staining pattern [28]. 

Based on the findings of the ToGA trial, current testing guidelines suggest that IHC should be the initial HER2 testing using validated assays. Samples with an equivocal IHC score of 2+ should undergo retesting using FISH or other in situ methods. Tests with 3+ overexpression by IHC or a FISH HER2–CEP17 ratio > 2.0 are considered positive, while IHC scores of 0–1+ are deemed HER2 negative [26]. 

Intra- and inter-tumoural heterogeneity of HER2 expression can cause inconsistent HER2 determination with a negative impact on treatment efficacy [29]. 

Beyond this, the high intraobserver variability, as shown in the VARIANZ study, especially in tumours with intermediate HER2 expression, demonstrates how interpreting HER2 amplification results can be challenging for pathologists requiring training and expertise [27].

Moreover, HER2 expression can also change upon progression of the disease. Previously established negative tumours can gain HER2 expression and, therefore, derive benefit from HER2 target therapies [30]. HER2 down-regulation is, on the other hand, one of the main mechanisms of acquired resistance [31]. Other mechanisms of resistance alteration in the HER2 receptor are the epithelial-to-mesenchymal transition [32], and Src-mediated activation of MAP/ERK and PI3K mTOR pathways [33]. 

Rebiopsies should be considered periodically or upon tumour progression to provide insight into mechanisms of resistance and targetable mutations. Longitudinal monitoring of the disease may be performed by implementing liquid biopsy in clinical practice [34,35].

## 3. HER2 Inhibition in Advanced GEA 

### 3.1. Historical Anti-HER2 Target Therapies in Advanced GEA

Several trials have investigated the safety and efficacy of different anti-HER2 drugs in advanced GEA. In the subsequent part of the review, clinical trials with the highest significant outcomes and clinical impact are described, and divided into each specific anti-HER2 targeted therapy (Table 1 and Figure 1).


*Trastuzumab*


Trastuzumab is the first monoclonal antibody developed against HER2. It binds to the extracellular domain IV of the receptor, inhibiting HER2 homodimerization and thereby preventing HER2-mediated signalling [36,37,38]. 

Since 2001, the addition of trastuzumab to cytotoxic chemotherapy has improved survival for patients with HER2-positive early or advanced breast cancer. In 2010, the ToGA (trastuzumab for gastric cancer) trial became the first prospective study to demonstrate the safety and efficacy of adding trastuzumab to chemotherapy in advanced GEA in a first-line setting [7]. 

This phase-3 multicenter randomised trial included HER2-positive metastatic GEA patients who received a combination of cisplatin plus fluorouracil (or capecitabine) every 3 weeks for six cycles or the same chemotherapy plus trastuzumab, followed by trastuzumab monotherapy. The primary endpoint of a significant increase in overall survival (OS) was met. Among the 584 patients included in the primary analysis, the mOS was 13.8 months in those who received trastuzumab plus chemotherapy, compared with 11.1 months for the control arm (HR 0.74; 95% CI 0.60–0.91; *p* = 0.0046). Median progression-free survival (mPFS) was 6.7 months in the trastuzumab plus chemotherapy group compared with 5.5 months in the chemotherapy-alone group (HR 0.71, 95% CI 0.59–0.85; *p* = 0.0002). The overall response rate (ORR) was 47% (5% complete, 42% partial) in the experimental arm and 35% (2% complete, 32% partial) in the control group. 

The recently published phase-3 KEYNOTE-811 trial evaluated the addition of the anti-PD1 pembrolizumab or a placebo to standard chemotherapy (fluoropyrimidine and platinum-based therapy) plus trastuzumab (every 3 weeks for up to 35 cycles) in the first-line treatment of advanced HER2-positive GEA [39]. Progression-free survival (PFS) and OS were the dual primary endpoints. After a median follow-up of more than 38 months in both arms, at the third interim analysis, the mPFS was 10.0 months vs.8.1 months (0.73, 95% CI 0.61–0.87; *p* = 0.0002) in the overall population. However, the mOS was not statistically significant (20.0 months vs.16.8 months; HR 0.84 [0.70–1.01]) and will continue to the final analysis. In the subgroup analysis according to PD-L1 expression, the benefit in PFS was not seen in patients not expressing PD-L1 (combined positive score CPS < 1). Based on this data, pembrolizumab in combination with trastuzumab, fluoropyrimidine, and platinum-based chemotherapy received a favourable recommendation from the EMA CHMP for the first-line treatment of HER2-positive GEA in tumours expressing PD-L1, with a CPS ≥ 1.

The INTEGA trial is a phase-2 study in which 88 patients were randomised to receive mFOLFOX6, trastuzumab, and nivolumab or a chemotherapy-free regimen with ipilimumab, nivolumab, and trastuzumab in a first-line setting [40]. Both arms were compared with the historical control of the ToGA trial. While considering the limitation of the small number of patients included, a better ORR (56%), mPFS (10.7 months), and mOS (21.8 months) were identified in the mFOLFOX6 plus nivolumab and trastuzumab group; this advantage was not seen in the chemo-free regimen. 

In a phase-2 trial, the combination of weekly trastuzumab with ramucirumab and paclitaxel was tested as a second-line treatment for HER2-positive GEA patients in an Asian population [41]. No dose-limiting toxicity was documented and, among the 50 patients enrolled, mPFS and mOS were 7.1 and 13.6 months, respectively. ORR was 54% and the disease control rate (DCR) was 96%, suggesting a safety profile of this combination in a pretreated HER2-positive population. 


*Pertuzumab*


Pertuzumab is an anti-HER2 humanised monoclonal antibody that binds to the extracellular dimerisation domain II, leading to the blockage of the HER2-mediated downstream pathway [42]. 

In the phase-3 JACOB trial, 780 HER2-positive metastatic GEA patients were randomised to receive pertuzumab (840 mg intravenously) or placebo plus trastuzumab with a platinum-based chemotherapy every 3 weeks as first-line treatment [43]. The mOSs were 17.5 months and 14.2 months in the pertuzumab arm and in the control arm, respectively (HR 0.84, *p* = 0.057). Despite the increasing trend in OS, the primary endpoint of the trial was not achieved.


*Lapatinib*


Lapatinib is a dual HER2/HER1 tyrosine kinase inhibitor that prevents the autophosphorylation of the receptor after ligand binding by blocking the intracellular domain [44].

In a first-line setting, the TRIO-013/LOGiC trial evaluated the efficacy of lapatinib plus CAPOX compared to chemotherapy alone in both Asian and Western populations of advanced GEA patients [45]. In the intention-to-treat population of 487 patients with centrally confirmed HER2-positive disease, the primary endpoint of OS was non-statistically significant, with 12.2 months in the lapatinib group and 10.5 months in the chemotherapy-alone group (HR 0.91, *p* = 0.392).

The TyTAN study tested the efficacy and safety of lapatinib in pretreated advanced HER2-positive GEA patients in an Asian population [46]. Patients were randomised to receive lapatinib plus weekly paclitaxel versus paclitaxel in monotherapy. Of the 261 enrolled, only 6% were previously treated with a trastuzumab-based regimen, and the OS primary endpoint was not statistically significant (11.0 months with lapatinib plus paclitaxel vs.8.9 months with chemotherapy alone [HR 0.84, *p* = 0.1044]). 


*T-DM1*


Trastuzumab emtansine (T-DM1) is an antibody–drug conjugate (ADC) composed of emtansine, a potent micro-tubular inhibitor, connected to the anti-HER2 monoclonal antibody trastuzumab [47]. 

The phase 2/3 GATSBY study evaluated T-DM1 in metastatic, previously treated, HER2-positive GEA [48]. In the phase-2 part of the trial, patients were randomised 2:2:1 to receive trastuzumab emtansine 3.6 mg/kg every 3 weeks (70 patients), trastuzumab emtansine 2.4 mg/kg weekly (75 patients), or a taxane (docetaxel or paclitaxel, 37 patients). After the pre-planned interim analysis, an additional 153 patients were randomly assigned to receive trastuzumab emtansine 2.4 mg/kg weekly, and another 80 patients received a taxane. Despite the impressive results of T-DM1 in breast cancer [49], the mOS (primary endpoint) did not show improvement, with 7.9 months in the T-DM1 arm and 8.6 months in the taxane arm (HR 1.15, *p* = 0.86).


*Trastuzumab deruxtecan (T-DXd)*


T-DXs is a novel HER2-directed antibody and topoisomerase I inhibitor conjugate. Upon release, the membrane-permeable payload causes DNA damage and cell death, resulting in the destruction of the targeted tumour cells and neighbouring cells, regardless of their HER2 expression. This bystander antitumor activity is crucial in cancers with heterogeneous HER2 expression, such as gastric disease [50,51]. 

The DESTINY-Gastric01 trial evaluated the safety and efficacy of T-DXd (6.4 mg/kg) compared to the physician’s choice of chemotherapy in an Asian population of HER2-positive GEA who had received at least two previous therapies, including trastuzumab [52]. Of 187 randomised patients, 125 received T-DXd and 62 chemotherapy (55 irinotecan and 7 paclitaxel). An ORR (primary endpoint) was reported in 51% of the patients in the T-DXd arm compared to 14% in the physician’s choice group (*p* < 0.001). The estimated mOS (secondary endpoint) was longer with T-DXd than with chemotherapy (12.5 vs. 8.4 months; HR 0.59; *p* = 0.01). 

**Table 1 ijms-25-03876-t001:** Landmark phase 2/3 clinical trials of anti-HER2 target therapies in advanced GEA.

Study (Name, Author, Year)	Number of Patients	Study Type	Line-Therapy	Drug	Treatment	Primary Endpoints
ToGA, Bang et al., 2010 [7]	594	Phase 3	First	Trast	Cape or 5FU + CDDP +/− trast	mOS 13.8 vs.11.1 ms (HR 0.74; *p* = 0.0046)
KEYNOTE-811, Janjigian et al., 2023 [39]	698	Phase 3	First	Trast	Trast + fluoropyrimidine and platinum-based therapy +/− pembro	mPFS 10.0 vs.8.1 ms (0.73, *p* = 0.0002) mOS 20.0 vs.16.8 ms (HR 0·84 [0.70–1.01]), not met
INTEGA, Stein et al., 2022 [40]	97	Phase 2	First	Trast	mFOLFOX6 + trast + nivo vs.ipi + nivo + trast	12-ms OS > 70% mFOLFOX6 + trast + nivo: 70% ipi + nivo + trast: 57%
Kim et al., 2023 [41]	50	Phase 1b/2	Second	Trast	Trast + pacli + ramu	mPFS 7.1 ms
JACOB, Tabernero J et al., 2018 [43]	780	Phase 3	First	Pert	trast + CDDP + cape or 5FU +/− pert	mOS 17.5 vs.14.2 ms (HR 0.84, *p* = 0.057)
TRIO-013/LOGiC, Hecht JR, 2016 [45]	545	Phase 3	First	Lap	CAPOX +/− lap	mOS 12.2 vs.10.5 ms (HR 0.91, *p* = 0.392)
TyTAN, Satoh T et al., 2014 [46]	261	Phase 3	Second	Lap	Pacli +/− lap	mOS 11.0 vs.8.9 ms (HR 0.84, *p* = 0.1044)
GATSBY, Thuss-Patience PC et al., 2017 [48]	415	Phase 2/3	Second	T-DM1	T-DM1 3.6 mg/kg vs.T-DM1 2.4 mg/kg vs.taxane (doce or pacli)	mOS T-DM1 2.4 mg/kg 7.9 vs.8.6 mstaxane (HR 1.15, *p* = 0.86)
DESTINY-Gastric01, Shitara et al., 2020 [52]	187	Phase 2	Third	T-DXd	T-DXd 6.4 mg/kg vs.CT (irinotecan, pacli)	ORR 51% vs.14% (*p* < 0.001)
DESTINY-Gastric02, Van Cutsem et al., 2023 [4]	79	Phase 2	Second	T-DXd	T-DXd 6.4 mg/kg	ORR 38%

Legend: 5FU: 5-fluorouracil; Cape: capecitabine; CAPOX: capecitabine, oxaliplatin; CDDP: cisplatin; CT: chemotherapy; doce: docetaxel; ipi: ipilimumab; lap: lapatinib; mFOLFOX6: modified FOLFOX (5-fluorouracil, oxaliplatin); mOS: median overall survival; mPFS: median progression-free survival; ms: months; nivo: nivolumab; ORR: overall response rate; pacli: paclitaxel; pembro: pembrolizumab; pert: pertuzumab; ramu: ramucirumab; T-DM1: trastuzumab emtansine; T-DXd: trastuzumab deruxtecan; Trast: trastuzumab.

The phase 2 DESTINY-Gastric02 study aimed to explore T-DXd in metastatic HER2-positive GEA patients with progressive disease during or after a first-line treatment with a trastuzumab-based regimen living in Europe and the USA [4]. Of the 79 enrolled subjects, the confirmed objective response was reported in 30 patients (38%), including three (4%) complete responses and 27 (34%) partial responses. Based on these clinically meaningful outcomes, the EMA CHMP adopted a positive opinion recommending the use of T-DXd in adult patients with metastatic HER2-positive GEA who have received a prior regimen with trastuzumab. Promising results are expected from the ongoing phase-3 DESTINY-Gastric04 trial (NCT04704934), which compares T-DXd with the second-line standard treatment of paclitaxel plus ramucirumab.

The DESTINY-Gastric03 is an ongoing phase 1b/2 study that evaluates T-DXd combined with chemotherapy and/or immune-checkpoint inhibitors (pembrolizumab or durvalumab) in first- or second-line treatments [53]. The preliminary results suggest tolerability and feasibility of T-DXd + 5-fluorouracil or capecitabine and the ORR obtained with T-DXd plus capecitabine (7/10 patients) showed promising activity in heavily pretreated, trastuzumab- and fluoropyrimidine-refractory GEA.

Based on breast cancer findings [54], the efficacy of T-DXd in HER2 low tumours was explored in the DESTINY-Gastric01 trial (cohort 1 included 19 patients with a central confirmed IHC 2+/ISH–negative HER2 status and cohort 2 included 21 patients with IHC 1+ status) [55]. The confirmed ORR was 26.3% in cohort 1 and 9.5% in cohort 2. Although preliminary, this study provides evidence of possible clinical activity of T-DXd in patients with heavily pretreated HER2-low GEA, but further data are mandatory.

### 3.2. Novel Anti-HER2 Target Therapies in Advanced GEA

Zanidatamab is a bispecific monoclonal antibody targeting the extracellular domains IV and II of HER2 [56]. In a first-line setting, a phase-2 trial of zanidatamab plus chemotherapy (mFOLFOX6, CAPOX or FP) showed a confirmed ORR of 79% and a DCR of 92%, with a median duration of response (DOR) of 20.4 months [57]. The confirmatory phase-3 study (HERIZON-GEA-01; NCT05152147), evaluating the combination of zanidatamab and chemotherapy with or without the PD-1 inhibitor tislelizumab, is currently enrolling.

Margetuximab is an Fc-engineered anti-HER2 monoclonal antibody [58]. In previously treated HER2-positive GEA, the phase-1b/2 CP-MGAH22-05 trial tested the combination of margetuximab plus pembrolizumab. The exploratory ORR was 18%, while the antitumor effect was greater in HER2 3+ and PD-L1-positive (CPS ≥ 1) tumours, with an ORR of 44% [59]. 

In a first-line setting, cohort A of the phase-2/3 MAHOGANY study explored the efficacy of margetuximab with the anti-PD-1 retifanlimab in HER2-positive (IHC 3+), PD-L1-positive (CPS ≥ 1), and non-microsatellite instability-high GEA [60]. The confirmed ORR (primary endpoint) was 53%, with a median DOR of 10.3 months and a DCR of 73%. The ORR observed in this study compares favourably to the ORR obtained with other chemotherapy-free approaches. Cohort B of the trial is currently ongoing, with the aim of enrolling HER2-positive patients, regardless of the PD-L1 status, and testing the safety and efficacy of margetuximab plus retifanlimab or the anti-PD1 and anti-LAG3 tebotelimab with chemotherapy [61]. 

Disitamab vedotin is an ADC composed of trastuzumab and the micro-tubular inhibitor auristatin E [62]. In an Asian population of 125 GEA patients pretreated with at least two previous lines, dasitamab vedotin showed an ORR of 24.8% and an mPFS and mOS of 4.1 and 7.9 months, respectively [63]. To support these preliminary findings, further data from large randomised trials, including a Western population, are required.

## 4. HER2 Inhibition in Locally Advanced GEA 

While the definition of HER2 overexpression and/or amplification is indispensable to establish the treatment strategy for advanced GEA, as well as PDL1, microsatellite status, and, in the near future, claudine 18.2 and fibroblast growth factor receptor (FGFR). The role of HER2-targeted therapies still needs to be defined in locally advanced disease [64]. 

Recently published ESMO guidelines [65] do not stratify patients by HER2 status to define the optimal perioperative treatment, nor do the NCCN ones [66].

A lot of clinical studies have evaluated the efficacy and tolerability of the association of trastuzumab and/or other HER2-targeted therapies to chemotherapy (CT) or chemo-radiotherapy (CTRT) for the curative treatment of GEA [67] (Table 2).

In the phase-II non-randomised HERFLOT trial the association of trastuzumab with 5-fluorouracil, oxaliplatin, and docetaxel (FLOT) was shown to be safe as a perioperative strategy in patients with locally advanced GEA. The pathologic complete response (pCR) rate was 21.4% (centrally assessed). A further 25% of patients had a near-complete response. The OS rate was 82.1% at 3 years while the median disease-free survival (mDFS) was 42.5 months [68]. Similarly, the association of XELOX and trastuzumab proved its feasibility and safety as a perioperative treatment in the Spanish phase-II NEOHX trial with an encouraging mOS of 79.9 ms and a 60 ms OS of 58%, although pCR was only 9.6% [69].

In the phase-III RTOG 1010, patients were randomised to radiotherapy plus carboplatin and paclitaxel (according to CROSS regimen) or the same treatment plus trastuzumab. No benefits in terms of pCR or DFS were observed [70]. 

These results demonstrate that the addition of trastuzumab to chemo(radio)therapy is feasible with no unexpected safety findings or increased treatment-related toxicities. 

Three phase-2 studies investigated the association of pertuzumab and trastuzumab with the standard treatment of locally advanced GEA. 

PETRARCA was a multicenter, randomised, trial study comparing FLOT vs.FLOT + trastuzumab and pertuzumab in locally advanced gastric cancer (>cT2 or cN+) [71]. This investigator initiated the clinical study, meant to be a phase II/III, and was closed after the presentation of negative results from the JACOB trial. Eighty patients were enrolled out of 100 planned. The addition of HER2 double blockage to FLOT significantly improved the pCR (35% vs.12%; p.002) and the rate of pathological lymph node negativity. The surgical morbidity and radical resection (R0) rates were similar. However, more grade > 3 events were observed, such as leukopenia (23% vs.13%) and diarrhoea (41% vs.5%). mDFS was 26 months in the FLOT arm vs. not reached in the trastuzumab–pertuzumab + FLOT arm (HR 0.58, 95% CI 0.28–1.19, *p* = 0.130); 2 years DFS were 54% and 70%, respectively. At the time of analysis, mOS had not been reached yet.

The TRAP study explored the association of trastuzumab and pertuzumab with CTRT (CROSS regimen) [72]. The primary end point (≥80% of patients completing the treatment with trastuzumab and pertuzumab) was met with 33 (83%) out of the 40 enrolled patients receiving all the planned cycles. All the patients undergoing surgery were radically resected. The pCR was 34%. At three years, PFS was 57% and OS was 71%, which favourably compared to a propensity-score-matched cohort treated with neoadjuvant CRT.

INNOVATION was a phase-II randomised trial [73]. Enrolled patients were randomised to neoadjuvant CT (FLOT, XELOX, or FOLFOX in European countries; cisplatin and capecitabine in Asian countries) for 3–4 cycles before and after surgery vs. the same regimen + trastuzumab or trastuzumab and pertuzumab. Anti-HER2 therapy was continued beyond CT for a total of 17 cycles. The study was closed due to slow accrual after 172 out of 215 patients were enrolled. The results of the major pathological response, the primary endpoint, were presented at the ASCO 2023 meeting. When all data were pooled, the response rate (RR) was 26.4%, 37%, and 23% in the CT, CT + trastuzumab, and CT + trastuzumab and pertuzumab groups, respectively. When only data from the FLOT/FOLFOX/XELOX studies were considered, the RRs were 37.9%, 53%, and 33%, respectively. The prespecified criteria of efficacy were not met (i.e., an improvement in RR from 25% with CT to 45% with CT + trastuzumab and pertuzumab). However, according to the authors’ conclusions, promising response rates were obtained with CT + trastuzumab, particularly with FLOT as the CT backbone. 

The safety of adding Lapatinib to perioperative treatment with an ECX regimen [74] and to neoadjuvant CTRT [75] was evaluated in two phase-II studies. Although the addition of lapatinib did not compromise operative management in both studies, it increased significantly the incidence of grade-3 diarrhoea when associated with ECX (from 0% to 24%, which was not considered safe by the authors) [74]. The CTRT study was closed after 12 patients were enrolled because of slow accrual, thereby limiting accurate analysis of the efficacy of this combination. Similarly, the efficacy of Sapitinib, an HER3, HER2, and EGFR inhibitor [76], was evaluated in a phase-II trial. Even if its safety profile proved to be acceptable with neoadjuvant CT, the sample size of the study was too small to draw conclusions on the efficacy of this combination [77].

**Table 2 ijms-25-03876-t002:** Landmark clinical trials of anti-HER2 target therapies in locally advanced GEA.

Study (Name, Author, Year)	Number of Patients	Study Type	Setting	Drug	Treatment	Response	Survival Outcomes
HERFLOT Hofheinz et al., 2021 [68]	56	Phase 2	Perioperative	Trast	FLOT + TRAST	pCR 21%	mDFS 42.5 ms; 3-year OS 82.1%
NEOHX, Rivera et al., 2021 [69]	36	Phase 2	Perioperative	Trast	XELOX + TRAST	pCR 9.6%	18 ms DFS 71%; mOS 79.9 ms
PETRARCA, Hofheinz et al., 2022 [71]	41 40	Phase 2 RCT	Perioperative	Trast + Pert	FLOT FLOT + Trast + Pert	pCR12% pCR 35%	mDFS not reached vs. 26 months (HR 0.58, *p* = 0.130)
INNOVATION, Wagner et al., 2023 [73]	35 67 70	Phase 2	Perioperative	Trast Trast + Pert	CT CT + Trast CT + Trast + Pert	pCR 26.4% pCR 37% pCR 23%	NR
Smyth et al., 2019 [74]	24 22	Phase 2	Perioperative	Lap	ECX ECX + Lap	TRG 1–2 9% TRG 1–2 25%	NR
Thomas et al., 2020 [77]	19 20	phase I/II	Perioperative	sapitinib	XELOX XELOX + sapitinib	NR	2 ys OS 90%, 6 ms DFS 100% 2 ys OS 64%, 6 ms DFS 85%
TRAP Stroes et al., 2020 [72]	40	phase II	Neoadjuvant	Trast + Pert	CROSS + RT + Trast + Pert	pCR 34%	2 ys DFS 70%
NRG Oncology/RTOG 1010, Safran et al., 2022 [70]	101 102	phase III RCT	Neoadjuvant	Trast	CROSS + RT 50.4 Gy CROSS + RT 50.4 Gy + Trast	29% 27%	mDFS 14.2 vs.19.6 ms (ns)
Shepard et al., 2017 [75]	12	phase II	Neoadjuvant	Lap	5FU + OX + RT 50 Gy + lapatinib	pCR 25%	NR

Legend: 5FU: 5 fluorouracil; CROSS = carboplatin + paclitaxel; ECX: epirubicin, cisplatin, capecitabin; FLOT: 5 fluorouracil, leucovorin, oxaliplatin, docetaxel; lap: lapatinib; mDFS: median disease-free survival, mOS: median overall survival; Neoadj: neoadjuvant; NR: not reported; OX = oxaliplatin; pCR: pathological complete response; pert: pertuzumab; RCT; randomised controlled trial; RT: radiotherapy; TRG: tumour regression grade; trast: trastuzumab; XELOX: capecitabine + oxaliplatin.

The positive results of DESTINY-Gastric01 and DESTINY-Gastric02 [4], which evaluated T-DXd in an advanced setting, have revitalised the interest in the use of antiHER2 agents other than trastuzumab. The ongoing phase-II EPOC2003 study aims to evaluate T-DXd in the neoadjuvant setting.

Many studies exploring the combination of HER2 target therapies plus immune-checkpoint inhibitors are ongoing. The initial results of a phase-II study evaluating the association of camrelizumab and trastuzumab with XELOX were presented at ASCO GI 2022. Out of 16 patients completing neoadjuvant treatment and undergoing D2 resection, 9 (56.3%) achieved a major pathological response, including 5 (31.3%) with pCR. The association of atezolizumab, XELOX, and trastuzumab in the perioperative setting is being investigated in a phase-II RCT (NCT04661150). In the adjuvant setting, patients with positive circulating tumour DNA (ctDNA) after resection will be randomised to trastuzumab or trastuzumab plus pembrolizumab; ct-DNA clearance will be the primary endpoint (NCT04510285). 

## 5. Future Perspectives

### 5.1. Chimeric Antigen Receptor T (CAR-T) Cell Therapy

CAR-T cell therapy is a new class of cellular immunotherapy. Patients’ autologous T-cells are modified ex vivo to incorporate engineered receptors specific for tumour antigens. Once these new T cells are reinfused, they activate an effective and specific antitumor immune response [78]. The heterogeneity of gastric cancer cells has limited the development of CAR-T cells in the treatment of GEA. Although HER2-targeted CAR-Ts have been successfully explored in other tumours, the evidence in GEA is only preclinical [79].

Song and colleagues evaluated the activity in vitro and in vivo of genetically modified human lymphocytes T expressing HER-2-specific CAR. HER 2-positive gastric cancer cells were effectively killed by these T cells in vitro. Moreover, an antitumor activity was also observed in HER2-positive gastric cancer xenografts during in vivo experiments. Finally, the engineered T cells inhibited the sphere-forming ability of HER2-positive gastric cancer cells derived from patients [80].

The safety and tolerability of CCT303-406 CAR-modified autologous T cells are currently being evaluated in a phase-I study in patients with pretreated stage-IV metastatic HER2-positive solid tumours relapsed or refractory to standard treatment, including GEA (NCT04511871).

### 5.2. Circulating Tumour DNA (ct-DNA) and Circulating Tumour Cells (CTCs)

Over the last few decades, several studies have highlighted the potential of CTCs and ct-DNA as a biomarker for the diagnosis, prognosis, and treatment monitoring of various cancers, including GEA [34,35,81]. However, up to these days, liquid biopsy is not part of everyday clinical practice, and it is not used to direct treatment decisions outside clinical trials.


*Ct-DNA*


In HER2-positive GEA, tissue biopsies display a high intra-tumoural heterogeneity in HER2 expression, which may prevent a correct diagnosis by the pathologist. The use of a plasma-based approach to detect HER2 expression represents a valid alternative. However, the concordance rate between HER2 amplification on tissue biopsy and ct-DNA ranged from 61% [82] to 91.2% [83] in two different studies.

Ct-DNA analysis could also predict and monitor the response to treatment with specific anti-HER2 agents. Wang and colleagues demonstrated that, in patients with HER2-positive advanced GEA receiving trastuzumab, HER2 copy changes detected in ct-DNA correlated with tumour response and anticipated modification in imaging exams [83]. In the phase-2 DESTINY-Gastric 01 trial, a baseline ct-DNA ERBB2 copy number above 6.0 was predictive of a greater radiological response rate to T-DXd compared to a copy number below 6.0 (ORR 75.8% vs.40.8%) [84]. Dedicated and prospective trials are needed to further explore and validate the predictive role of HER2 amplification detected in ct-DNA, as well as of the optimal amplification level cut-off.

Ct-DNA analysis may also be used to investigate mechanisms of resistance to anti-HER2 agents. Next-generation-sequencing (NGS) ct-DNA detection was performed on a small group of 14 patients to detect the molecular alterations associated with resistance after disease progression to first-line treatment with trastuzumab [82]. HER2 amplification was not detectable anymore in 73% of patients, while in those with persistence of HER2 amplification, BRAF, KRAS, and PIK3CA mutations were identified as potential molecular alterations causing resistance to trastuzumab. Mutations of NF1 and MET amplification, even if HER2 amplification was maintained, were identified as other potential mechanisms of resistance in a small study of 17 patients progressing during trastuzumab treatment. [85].


*CTCs*


Mishima and colleagues recently demonstrated that, in metastatic HER2-negative GEA (as assessed by tumour biopsy), HER2 amplification can be found in CTCs [86]. The disease outcomes of these patients, once treated with trastuzumab-based therapy, are comparable to those reported in the experimental arm of the ToGA trial.

Matsushita et al. confirmed the clinical benefit derived from treating with trastuzumab GEA with HER2 positivity detected by CTCs analysis despite a HER2-negative tumour tissue biopsy. According to multivariate analysis, HER2 expression on CTCs was an independent prognostic factor for both OS and PFS [87].

## 6. Conclusions

Since the publication of the TOGA trial in 2010, the association of platinum-based chemotherapy and trastuzumab has become the standard-of-care treatment of HER2-positive advanced GEA. However, other drugs such as lapatinib, pertuzumab, and T-DM1 failed to further improve the outcomes of HER2-positive GEA patients both in early and advanced settings, with no valid options available after disease progression on trastuzumab-based first-line treatment.

Recently, the ADC trastuzumab deruxtecan led to positive results in pretreated advanced HER2-positive GEA, thereby renewing the interest in preclinical and clinical research in this setting. Its unique mechanism of action (i.e., the bystander antitumour effect targeting cells regardless of HER2 expression) is highly effective in GEA, which frequently presents a heterogeneous expression of HER2. Trastuzumab deruxtecan is currently being evaluated in first-line treatment and earlier stages.

Among newer drugs, Margetuximab, an antiHER2 monoclonal antibody, and zanidatamab, a bispecific antibody, have shown promising results also in the first-line setting and are now been tested in wider populations and in earlier stages.

The role of another ADC, disatamab vedotin, which has shown efficacy in an Asian population, although promising, needs to be tested in larger trials, including also Western patients.

In GEA, identifying HER2 alterations is strongly recommended in clinical practice in order to activate specific anti-HER2 treatment. Indeed, HER2 alteration and, more recently, PD-L1 expression are the only predictive biomarkers of response that can be useful to a clinician in the first-line decision-making process. Based on a strong preclinical rationale, combining immune-checkpoint inhibitors with HER2-directed agents has catalysed the interest of clinical research in the treatment of HER2-positive GEA [88]. The positive results of KEYNOTE-811 have changed the standard of care for HER2-positive/PDL1-positive advanced GEA. After this, the combination of anti-HER2 treatments, and immunotherapy, including but not limited to PD/PDL1 and CTLA4 inhibitors, is being explored in metastatic and locally advanced disease, even with chemo-free approaches such as in the phase-2/3 MAHOGANY study.

CAR T-cell development is still in a preclinical setting in GEA and needs to be tested on large patient populations before entering clinical practice. Beyond efficacy, the CAR T side-effects profile (systemic and local) needs to be carefully considered. It requires a meticulous balancing of CAR T-cell activation to stimulate cytokine secretion but avoid cytokine storm. Moreover, considering the heterogeneity of HER2 expression and the immunosuppressive tumour microenvironment, genetic research should focus on designing CAR T cells with enhanced anti-tumour activity and ability to target multiple antigens, and on reversing the immunosuppressive tumour microenvironment, for example by associating PD-L1 or PD-1 inhibitors [79].

The application of liquid biopsy in clinical practice will allow a continuous monitoring of disease status, both for locally advanced/resectable and for metastatic/unresectable tumours. Detection of minimal residual disease (MDR) after surgery, early diagnosis of relapse before radiological evidence, identification of acquired resistance, and molecular (re)profiling during treatment are some of the most promising applications that will allow for optimising the treatment results of HER2-positive GEA.

Challenges to overcome in the future include the identification of predictive biomarkers capable of selecting patients who may derive benefit from a defined treatment, the development and implementation of assays to allow real-time assessments and reassessments of the disease and monitoring the efficacy of HER2 treatments, and the identification of new treatment strategies to further improve the outcomes of HER2-positive GEA.

## Figures and Tables

**Figure 1 ijms-25-03876-f001:**
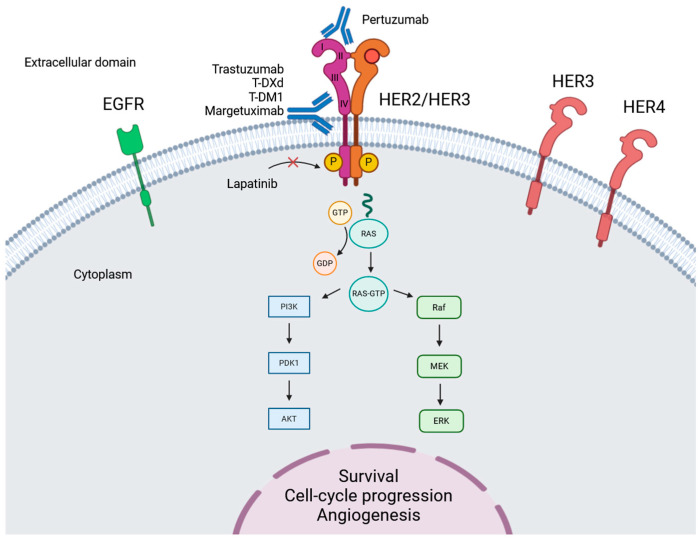
The figure represents the HER2 signalling pathway and the extra- and intra-cellular binding site of anti-HER2 drugs. HER2 receptor is composed of four extracellular domains identified as I, III, III, and IV.

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
