# Peer review of "Targeting HER2 in Gastroesophageal Adenocarcinoma: Molecular Features and Updates in Clinical Practice"

_ijms, 2024, doi:10.3390/ijms25073876_

Round 1

Reviewer 1 Report

Comments and Suggestions for Authors

1. Please re-do the titling of all drugs discussed in section 3.1

2. The conclusion could include one or two most promising new drugs for different contexts for advanced and locally advanced GEAs.

3. Also would be add which of the future perspectives listed (CAR-t/ct-DNA/CTCs) would be more beneficial in GEA context. 

The manuscript in its current form fails to add understanding of significant advances in targeting HER2 in GEA. It would be good to discuss if HER2 targeting is in fact worth pursuing and  to predict which of the therapies from a molecular targeting perspective and current clinical evidence would be most beneficial. 

Comments on the Quality of English Language

Minor typos and grammatical restructuring necessary

Author Response

Dear Colleague

thank you for all your comments and suggestions. We have carefully considered them and modified our manuscript.

Here are replies to your comments:

  • titling was redone rewriting the names of the drugs in italics, we did not add subparagraphs numbers on purpose (otherwise the numbering would have been 3.1.1, 3.1.2…adding unnecessary complexity to the manuscript)
  • we modified the conclusions according to your suggestions. In particular, we put emphasis on the most promising drugs in advanced and locally advanced GEA, we added the possible applications of ct-DNA in clinical practice and underlined that the development of CAR-T cells, although with a great potential, is still at the beginning for GEA. The role of the association of HER2 target agents and immunotherapy was also discussed as the most promising combination to explore in the future in this setting.

Reviewer 2 Report

Comments and Suggestions for Authors

Dear colleagues,

The manuscript “Targeting HER2 in Gastroesophageal Adenocarcinoma: Molecular Features and Updates in Clinical Practice” by Bonomi et al., is an interesting review upon HER-2 positive gastroesophageal cancer current and future therapeutic landscape. The work clearly discusses molecular background and main impactful trials over the time, and I think is worth publishing.

However, I would like to address some minor remarks:

1.       Please, check English editing. Also, abbreviations should be consistent throughout the text.

I suggest some corrections:

- (Line 28) “are” not “is”

- (Line 76 - Figure 1) “represent-s”

- (Line 77) “four” instead of “IV”

- (Line 171) I suggest to add “While’ considering the limitation….”

- (Line 281) I believe is “Disitamab vedotin”. Also correct further in text (Line 443)

- (Line 343) Pleas clarify RR abbreviation

- (Line 364) Please correct “…the combination of HE2 target therapies…”

- (Legend Table 1) Some abbreviations are missing (i.e. pert, pacli)

2.       Table 1 and 2: I suggest to adapt the two and list clinical trials in chronological order. Format need to be improved.

3.       (Line 88) “In GEA HER2 overexpression is heterogeneous and may vary based on the primary tumour and distant metastases, as well as the primary location of the cancer and its histology…”. Please clarify.

4.       (Line 161-162) About KEYNOTE-811, I believe that data reported on mPFS refer to second interim analysis. I also suggest to mention that mPFS  did not differ in the population with a PD-L1 CPS of less than 1, in order to better motivate EMA recommendation. Also, mOS in control group was 16.8 months, please correct.

Comments on the Quality of English Language

Minor editing of English language required

Author Response

Dear Colleague

thank you for your comments and suggestions. We have carefully considered them and modified our manuscript.

1) all the corrections you suggested in point 1 were done, english editing was revised and all the abbreviations reported in the manuscript were double checked to guarantee consistency;

2) table 1 and 2 were adapted and trials evaluating the same drug were listed in chronological order (starting from trastuzumab);

3) line 88 was rewritten. HER2 heterogeneity (i.e. variability in HER2 staining within the same biopsy) was addressed later in our manuscript, so we deleted that part to avoid repeating the same concepts;

4) we modified KEYNOTE-811 results with the third interim analysis and we added a sentence about PD-L1 negative subgroups as you suggested.

Kind regards.

Reviewer 3 Report

Comments and Suggestions for Authors

Authors present in this article a comprehensive review of the clinical advancements in targeted therapy and immunotherapy for locally advanced and advanced Gastro esophageal adenocarcinoma. They summarized the molecular classification and recapitulate the history of different molecules used in clinical practice and dress future perspectives. The review article is well structured and easy to read. Although the subject is well known it is still in development and many results of ongoing trials are not yet available and the authors list all the update or available information around these trials.

no particular comments.

Author Response

Dear colleague,

thank you  for your comments, you’ll find a slightly modified version of the manuscript you approved according to the comments made by other reviewers.

Kind regards.

The authors

Round 2

Reviewer 1 Report

Comments and Suggestions for Authors

Thank you for the comments. The manuscript looks ready apart from some grammatical errors/sentence restructuring.

Comments on the Quality of English Language

Needs some sentence rewriting. Will be helpful if you upload a version without the edits tracking and the comments column on the right.